# Graphene Oxide and Polymer Humidity Micro-Sensors Prepared by Carbon Beam Writing

**DOI:** 10.3390/polym15051066

**Published:** 2023-02-21

**Authors:** Petr Malinský, Oleksander Romanenko, Vladimír Havránek, Mariapompea Cutroneo, Josef Novák, Eva Štěpanovská, Romana Mikšová, Petr Marvan, Vlastimil Mazánek, Zdeněk Sofer, Anna Macková

**Affiliations:** 1Institute of Nuclear Physics of CAS, v.v.i., Husinec-Rez, 250 68 Rez, Czech Republic; 2Department of Physics, Faculty of Science, University of J.E.Purkyně, 400 96 Usti nad Labem, Czech Republic; 3Department of Inorganic Chemistry, University of Chemistry and Technology, 166 28 Prague 6, Czech Republic

**Keywords:** graphene oxide, polymers, carbon ion micro-beam writing, humidity sensors

## Abstract

In this study, novel flexible micro-scale humidity sensors were directly fabricated in graphene oxide (GO) and polyimide (PI) using ion beam writing without any further modifications, and then successfully tested in an atmospheric chamber. Two low fluences (3.75 × 10^14^ cm^−2^ and 5.625 × 10^14^ cm^−2^) of carbon ions with an energy of 5 MeV were used, and structural changes in the irradiated materials were expected. The shape and structure of prepared micro-sensors were studied using scanning electron microscopy (SEM). The structural and compositional changes in the irradiated area were characterized using micro-Raman spectroscopy, X-ray photoelectron spectroscopy (XPS), Rutherford back-scattering spectroscopy (RBS), energy-dispersive X-ray spectroscopy (EDS), and elastic recoil detection analysis (ERDA) spectroscopy. The sensing performance was tested at a relative humidity (RH) ranging from 5% to 60%, where the electrical conductivity of PI varied by three orders of magnitude, and the electrical capacitance of GO varied in the order of pico-farads. In addition, the PI sensor has proven long-term sensing stability in air. We demonstrated a novel method of ion micro-beam writing to prepare flexible micro-sensors that function over a wide range of humidity and have good sensitivity and great potential for widespread applications.

## 1. Introduction

Humidity is an important environmental characteristic that affects physical, chemical, and biological processes; therefore, the control of humidity, which causes material corrosion, is essential for almost all human health, activities, and comfort [1]. To achieve this, we must rigorously monitor and control humidity using a method with a high sensitivity, broad detection range, quick response, and short recovery time [2]. Humidity sensors, which are based on changes in physical or chemical properties caused by absorbed water molecules, can be non-hydrophilic or hydrophilic [1]. The most common types of humidity sensors are based on chemo-resistive interactions due to their simplicity, fast response, high sensitivity, and low cost [3]. The working principle of the chemo-resistive sensors is based on changes in electrical conductivity caused by the absorption of water molecules on the sensor surface sites. Interactions between the adsorbed water molecules and the oxygen ions modulate charge carrier concentration and subsequently alter electrical conductivity (or resistivity) [4,5]. Recently, various humidity sensors based on various sensing materials, such as metal oxides, silicon, ceramics, semiconductors, carbon, and transition metal dichalcogenides, have been developed and tested [2,6]. For example, most ceramics are not very sensitive to humidity levels below 20% [5]. Semiconductors are promising due to their high precision, but they only function well in high temperatures [5]. Another common type of moisture-sensing material is organic materials, such as polymers or, more recently, graphene or graphene oxide [5].

Organic solids, such as graphene oxide (GO) or polymers, are excellent candidates for the fabrication of chemo-resistive sensors, due to their detection ability at room temperature, excellent sensitivity, mechanical stability, short reaction time, low cost, commercial availability, easy processing, and flexibility [1,5,7,8]. The water molecules absorbed by hydrophobic polymers, such as polyimide, poly(methyl methacrylate), or polyethylene–terephthalate, change their relative permittivity [9]. On the other hand, the water molecules in hydrophilic materials, such as graphene oxide, are trapped and bonded on their structure and increase the dielectric constant and the capacitance [2,10]. Moreover, GO and porous polymers have extremely large specific surface areas, allowing for the significant absorption of the water molecules, and thus an excellent sensing performance. Due to their extremely large specific surface area of GO and porous polymers, almost all of their atoms are located on their surfaces, and thus able to interact with water molecules [11]. For instance, polyimide (PI) sensoric properties have been studied at a relative humidity between 0% and 85% due to the low hysteresis of PI, bi-directional water transport (absorption and evaporation), good linearity, and high sensitivity to humidity [12,13,14]. Additionally, poly(methyl methacrylate) has a high sensitivity to humidity, mechanical stability, and processability [15,16,17]. Humidity sensors based on GO drops or spray deposited on some conductive structures (carbon or metallic) were successfully tested at relative humidity (RH) and ranged from 15 to 95%. The sensitivity of some GO sensors was found to be more than 10 times higher than that of the best conventional sensors [2,9].

Although there are several methods, such as inkjet printing, laser writing, or metal sputtering, for the fabrication of in-plane organic humidity micro-sensors, ion-beam writing has not yet been used [18,19,20]. Ion-beam lithography is an efficient technique for the very local and precise modification of chemical and functional properties (patterning), which in the case of organic compounds, provides the possibility of choosing the resulting properties (degree of reduction, modification of sp^2^/sp^3^ ratio of carbon hybridization, sensory/catalytic properties, and dielectric properties). In addition, ion irradiation has several advantages, such as the absence of chemical agents, the absence of unwanted oxides formation, less formation of residual impurities, and relatively simple and cost-effective mass production [21,22]. Conducting polymers are important in the field of flexible sensing technologies, and a number of studies on the ion irradiation of polymers reported an increase in electrical conductivity caused by polymer carbonization [23]. When the fluence of ions during irradiation exceeds 10^13^ cm^−2^, the threshold for the overlapping of individual ion tracks is overcome, and the π-bonded carbon clusters increase in number, aggregate, and create a network of conjugated carbon bonds [23,24]. Thus, in the range above 5 × 10^13^ cm^−2^, the nucleation and growth of nano-sized carbon-enriched clusters are expected until a quasi-continuous carbon-embedded layer is formed [23]. The origin of the electrical conductivity of irradiated polymers is then considered to be the hopping or tunneling of electrons between the conductive carbon islands [25].

In this study, we focus on carbon ion beam writing for the maskless fabrication of micro-meter electrode systems to insulate organic matrices for use in humidity sensors. Graphene oxide (GO), polyethylene–terephthalate (PET), polyimide (PI), and poly(methyl methacrylate) (PMMA) foils were irradiated using a focused 5 MeV C^3+^ ion micro-scale beam to induce the carbonization/deoxygenation/dehydrogenation and locally increase electrical conductivity. The ion fluences of 3.75 × 10^14^ cm^−2^ (1800 nC/mm^2^) and 5.625 × 10^14^ cm^−2^ (2700 nC/mm^2^) were used, for which a significant increase in electrical conductivity was previously reported for GO, PI, and PMMA [26,27,28]. The irradiated part of the dielectrics is the electrode, and the non-irradiated part is the sensing area. The following topics are addressed: (i) the usefulness of a carbon ion beam with a specific electronic-to-nuclear stopping ratio for maskless microstructuring of the chosen materials; (ii) the structural modification differences among PMMA, PI, PET, and GO during carbon ion-beam irradiation; (iii) the obtainable quality of microstructures and their morphologies; (iv) electrical responses of created structures to different surrounding humidity levels and humidity sensing.

## 2. Materials and Methods

GO was prepared using a permanganate oxidation method (modified Hummers’ method) and subsequently separated using a centrifuge, as described in [29]. A GO film was filtrated on a polycarbonate membrane (Nucleopore, 0.45 µm pore size) and dried at 50 °C. The density of the as-prepared GO film (1.35 g·cm^−2^) was determined by weighing the accurately cut portion of the GO film. The 50 μm-thick foils of PMMA (density (ρ) = 1.19 g·cm^−3^, glass transition temperature (T_g_) = 105 °C, melting temperature (T_m_) = 160 °C, surface resistance (σ_e_) = 10^14^ Ohm/sq), PI—Kapton (ρ = 1.19 g·cm^−3^, T_g_ = 260 °C, T_m_ = 340 °C, σ_e_ = 10^16^ Ohm/sq), and PET—Mylar (ρ = 1.3 g·cm^−3^, T_g_ = 95 °C, T_m_ = 260 °C, σ_e_ = 10^14^ Ohm/sq) were supplied by Goodfellow, Ltd., Huntingdon, UK [30,31].

Ion-beam lithography was performed in an Oxford Microbeams micro-beam chamber utilizing 5 MeV C^3+^ with a 250 pA current. The carbon beam spot on the foil surface was focused to 4 × 15 µm^2^. The ion fluences were 3.75 × 10^14^ cm^−2^ (1800 nC/mm^2^) and 5.625 × 10^14^ cm^−2^ (2700 nC/mm^2^). The microscopic structure was prepared as an image of 900 × 900 pixels^2^ (pixel size of 1 µm^2^, see Figure 1). The line width was 10 μm, and the inter-line spacing was 20 μm. The overlapping line segments were 500 μm in length (see Figure 1). The total width of the produced micropattern was 880 μm. The same foils were irradiated throughout their area under the same irradiation conditions (5 MeV C^3+^ ions, ion fluences 3.75 × 10^14^ cm^−2^ and 5.625 × 10^14^ cm^−2^) as reference samples provided for the investigation by using RBS, ERDA, and XPS analyses, where appropriately broader modified sample area is required. The current of ions during irradiation was maintained at 10–15 nA·cm^−2^, significantly less than the current of ions used in microbeam writing (250 pA.60 µm^−2^–4.2 × 10^2^ µA·cm^−2^). The higher current could not be reached during broad beam irradiation using a Tandetron accelerator.

The compositional changes in the connection to carbon ion irradiation were studied using Rutherford backscattering spectrometry (RBS) and Elastic recoil detection analysis (ERDA). The RBS/ERDA spectra were measured using a He beam of energy at 2.0 MeV. An Ultra-Ortec PIPS detector registered the back-scattered ions in Cornell geometry with a scattering angle of 170°. The input angle of the initial ERDA beam was 75°, and the scattering angle was 30°. The recoiled particles were registered by a Canberra PIPS detector equipped with a 12 µm Mylar foil. The current of the He ions used in the RBS and ERDA analyses was ~5 nA. Several RBS spectra were measured at random locations in the beam to reduce the effect of sample degradation. The resulting spectrum is a sum of incremental spectra. The concentration of the elements was analyzed using SIMNRA software [32].

Raman spectroscopy was measured using an inVia Raman microscope (Renishaw, Wotton-under-Edge, England). The spectrometer operates using a backscattering geometry with a CCD detector and a 50 mW Nd-YAG 532 nm laser with a 50× magnification objective. The calibration of Raman spectrometer was performed with a silicon sample that provided a peak position at 520 cm^−1^. To minimize damage to the sample, no more than 5% of the overall laser power was used. For the measurement, the samples were cast on a silicon wafer from a suspension of isopropanol (1 mg·mL^−1^).

Scanning electron microscopy (SEM) images of the GO, PMMA, PI, and PET micro-structures were obtained using a field emission electron source (TescanLyra dual beam microscope, 10 kV). The sample was adhered to a carbon conductive strip for SEM measurement.

The X-ray photoelectron spectroscopy (XPS) was performed using the ESCAProbeP from Omicron Nanotechnology Ltd. (London, England), with exposed area dimensions of 2 × 3 mm^2^ that were analyzed. The X-ray source was monochromatized to 1486.7 eV, and each measurement was performed with a step of 0.05 eV. CasaXPS, ver. 2.3.24 software was used for spectral evaluations.

To examine the effect of humidity on the electrical properties of prepared structures directly associated with the ability to sense humidity, the modified foils were placed in an environmental chamber in which relative humidity (% RH) was controlled (RH range 5–60%) by mixing dry and wet air and monitored using a BME 280 pressure/humidity/temperature sensor (Bosch Sensortec, Reutlingen, Germany). The change in electrical capacitance with varying humidity levels was determined by the measurement of the frequency dependence of the high-pass filter output voltage with a constant current of 0.5 mA, see Figure 2. The resistance of the used resistor was 1 MΩ. The current waveforms (sinus) with a frequency range of 20–100 kHz were generated using a Keithley 6221 source, and AC voltage was recorded using an Owon PDS 7102T oscilloscope (Owontech, Zhangzhou, China) (see Figure 2). The resulting values were compared with those of ceramic commercial capacitors. The change in the electric resistance of prepared structures with variable humidity was measured using a standard 2-point measurement of sheet resistivity utilizing the Keithley 6317B Electrometer (Tektronix, Solon, OH, USA).

## 3. Results

### 3.1. Elemental Composition by RBS and ERDA

The element composition of non-modified and irradiated samples was determined using ion-beam spectroscopic methods (RBS and ERDA) with 2 MeV He ions. The available information depth of RBS with a He ion beam energy of 2 MeV in polymers and GO is approximately 1 µm, and it is approximately 0.5 µm for ERDA. The RBS and ERDA spectra from pristine and irradiated samples are shown in Figure 3. The atomic concentrations of C, H, and O and their ratios are presented in Table 1. The PI and GO samples show only minor compositional changes, i.e., slight hydrogen and oxygen depletion and a mild carbon concentration increase. After irradiation with a carbonion fluence of about 5.625 × 10^14^ cm^−2^, the C/O and C/H ratios in GO growth change from 3.6 to 4.4 and from 7.6 to 8.9, respectively. Additionally, only low GO carbonization occurs, and carbon concentration in GO increases by about 3 at. %. The C/O and C/H ratios increase from 4.4 to 6.8 and from 2.2 to 2.7, respectively, in PI. The irradiation with 5 MeV carbon ions with predominant electronic stopping leads to the weak bond scission in GO and PI and the release of low-mass gaseous fragments, followed by GO and PI reduction and carbonization [33]. A higher ion fluence does not have a more significant effect on GO and PI elemental composition compared to the lower fluence. The carbon irradiation of PET and PMMA polymers has the opposite effect on oxygen concentration compared to PI and GO. While in GO and PI, the oxygen concentration decreases, in PET and PMMA, the amount of oxygen after carbonion implantation increases. The hydrogen concentration in PET and PMMA decreases more significantly than in PI and GO. The H concentration decreases after implantation using carbonion fluence of 5.625 × 10^14^ cm^−2^ in PET from 36.3 to 25.4 at. % and in PMMA from 43 to 18 at. %. The increase in O concentration in PET and PMMA after ion irradiation may be caused by the post-irradiation oxidation of the damaged layer. Oxygen from the surrounding atmosphere is diffused into the open structures of polymers and captured in the defects [24]. Additionally, the significant release of hydrogen from PET and PMMA is caused by the cleavage of macromolecular chains and the formation of free radicals during irradiation, predominantly in the electronic stopping mode [34].

### 3.2. Surface Morphology by SEM

The changes in PET, GO, PI, and PMMA morphology caused by carbon ion irradiation, acquired by SEM, are shown in Figure 4. PI is known for its radiation and thermal durability caused by a high aromaticity, as cited above. Therefore, the PI surface is maintained after carbon irradiation to its nearly featureless morphology (see Figure 4c). Additionally, PET is an aromatic polymer, but its structure is simpler than that of PI; PET has one rigid benzene ring, and the radiation and thermal resistance of PET is not as high as PI [35]. The interface between irradiated and non-irradiated components on the PET surface (Figure 4b) is more significant compared to PI, and small grains are visible on the PET surface. The PET structure is susceptible to modifications by chain scission, which leads to chain shortening, a lower molecular weight, the escape of released oxygen and hydrogen-based molecules, and the formation of micro-cracks and small grains on the PET surface [36,37]. The geometry and dimensions of irradiated and non-irradiated parts on the surface of PET and PI confirm this assumption and the interface between irradiated and pristine parts is straight and without structural deviations. On the contrary, the PMMA after irradiation exhibits significant morphological alteration to destruction and raggedness (Figure 4d). These significant changes in the PMMA surface are associated with its low thermal stability and can be interpreted as the basis for ion-induced heating and melting by a high ion-current (~420 µA·cm^−2^ in our case), which causes local heating in the vicinity of the ion impact [38,39,40]. Moreover, the high fluence of MeV carbon ions, above 3.4 × 10^12^ cm^−2^, leads to the removal of pendant side chains in PMMA, and the creation of short chains with interrupted bonds and cross-linking. The shrinkage of the irradiated PMMA structure leads to surface cracking and swelling [41,42]. The GO surface (Figure 4a) typically consists of ~10–20 µm GO flakes and creates a platelet structure. The shape of the edges of the irradiated GO parts is not as sharp as in the case of PI and PET. The interface between unaffected and exposed GO is rippled and copies the shape of individual GO flakes. This may be due to the heat reduction in non-irradiated components adjacent to their irradiated counterparts.

### 3.3. Elemental Composition by EDS

Energy-dispersive spectroscopy (EDS), simultaneously measured with SEM provides a qualitative insight into differences in the elemental composition of irradiated and nonirradiated parts (see Figure 5). An EDS analysis performed with 10 keV electrons confirms similar changes in the elemental composition foils as those in RBS. Oxygen release is observed in GO and PI after ion irradiation and is more pronounced with the increase in ion fluence (Figure 5a–c,h–j). The completely opposite state is then observed in PET and PMMA samples (Figure 5e–g,k–m). The oxygen concentration increase after ion irradiation is caused by O diffusion into the samples from the ambient atmosphere. The EDS data clearly confirm a reduction in PI and GO induced by carbon irradiation. The nominal difference in O and C concentrations between RBS and EDS is associated with different principles. EDS probes a few microns from the sample surface and causes integral information. On the contrary, RBS has a depth resolution in the range of several nanometers and is more sensitive to the detection of changes in the irradiated layer. The trace amounts of sulfur in GO originate from GO film synthesis (see Figure 5a–c).

### 3.4. Structure Analysis by Raman Spectroscopy

Raman spectroscopy provides a detailed analysis of structural changes caused by carbon beam irradiation (see Figure 6). The pristine PI Raman spectrum is overwhelmed by a high fluorescence background due to the strong optical absorption and excitation of π-electrons in the phenyl rings (Figure 6c) [43]. The PI fluorescence background decreases after carbon ion irradiation, and two peaks at 1360 and 1580 cm^−1^ appear, whose intensity increases with increasing ion fluence. These two peaks correspond to the D and G broad bands of the disordered graphitic structure. The G band is connected to the E_2g_ Brillouin zone-center phonon density of states in the sp^2^-bonded graphite-like carbon; the D band is caused by defects in the momentum conversation and is ascribed to the point defects and effects on the edges [43,44]. The phenyl rings most likely break, and hydrogen release and disordered graphene-like sp^2^ structure growth are caused in the PI structure, leading to an increase in PI conductivity.

The Raman spectra of non-irradiated PET and PMMA show several peaks originating from PET and PMMA monomer structures (see Figure 6b,d). The pristine PMMA significant modes originate from C-C-C symmetric stretching (814 cm^−1^), C-C stretching (987 cm^−1^), C-H_2_ bending (1455 cm^−1^), C=C stretching vibration (1590 cm^−1^), C=O stretching (1730 cm^−1^), and C-H symmetric stretching (2800–3100 cm^−1^) [45]. The Raman spectrum of pristine PET comprises modes resulting from C-C-C in-plane and out-of-plane vibrations (632 and 704 cm^−1^), C-C breathing (857 cm^−1^), CH_2_ wag (1381 cm^−1^), benzene rings (1613 cm^−1^), and C=O vibration (1727 cm^−1^) [46]. After ion implantation, the D and G peaks grew in the PET and PMMA spectra, replacing all other spectral features of the pristine polymers and indicating the amorphization and partial carbonization of the irradiated structure, which contributes to the change in electrical properties.

The most prominent peaks are the D and G peaks in all Raman spectra of GO (Figure 6a). The G mode in Raman spectra of graphene-based samples manifests the motion of the in-plane bond stretching of sp^2^ carbon pairs, and the intensity of this signal represents the degree of the GO lattice arrangement [47]. The D mode is not present in pristine graphene due to crystal symmetries, and this D signal represents defects induced in GO and the effects of the structure of GO edges [48]. The next broad mode, connected to the vibration of non-benzene five-membered carbon rings, is around 1460 cm^−1^ [45]. The D-peak intensity decrease and 1460 cm^−1^ peak increase after carbon irradiation are more pronounced with increased ion fluence. The decrease in the D peak represents the reduction in the epoxide and hydroxyl parts of GO and the decline in multipeaks between 2400 and 3200 cm^−2^ is shown in the growth of sp^2^ domains [49,50], which was also reflected by XPS, RBS, and EDS. The peak at around 1460 cm^−1^ growth is caused by the formation of non-six-carbon rings [51]. It is evident that carbon irradiation leads to a reduction in GO and an increase in irregular graphene-like structure, which will increase the electrical conductivity of irradiated samples.

### 3.5. Surface Chemistry by XPS

X-ray photoelectron spectroscopy (XPS) describes the changes in surface chemistry, which at first, directly interacts with the environment surrounding the analyzed sample and significantly influences the final electric and sensory properties. The deconvolution of high-resolution C1s peak for non-modified samples as well as for samples irradiated with the highest ion fluence (5.625 × 10^14^ cm^−2^) are presented in Figure 7. The C1s peak of XPS spectra of pristine GO consists of four bonding states (Figure 7a): -COOH (289.16), -C-C (284.71 eV), -C-OH (285.9 eV), and -C=O (287.35 eV) [52]. The -C-C, -C-OH, and -C=O carbon hybridization states are very intense in the pristine GO and indicate the presence of oxygen functional groups (carbonyl, epoxy, hydroxyl) and a considerable degree of sample oxidation [27,43,53,54]. After ion irradiation (Figure 7e), the -C=O and -C-OH peak intensity in GO significantly decreases, which corresponds to the previously mentioned removal of oxygen-containing groups supplemented with the creation of new carbon groups. The deoxygenation of the GO surface layer enhances the sample’s electric properties and leads to a higher conductivity. Moreover, the increase in carboxyl COOH groups that mainly bind at the edges of the GO structure, indicates the rupturing of the GO structure and an increase in defects [55].

The deconvolution of the C1s region of the pristine PI XPS spectra (Figure 7c) displays five carbon hybridization: -C-C carbons from the aromatic rings of oxydianiline PI part (284.56 eV) and -C-C in benzene rings of pyromellitic dianhydride structure of PI (285.19 eV), single -C-OH and -C-N bonds (286.06 eV), -COOH at 288.51 eV, and finally C=O carbons double bonded in the imide ring (287.13 eV) [56,57]. Only three signals were left in the C1s PI spectra after ion irradiation (Figure 7g). The -C-OH and -C=O peaks completely disappeared, and this effect was accompanied by the -C-C and -COOH intensity growth. The decreasing abundance of oxygen-containing bonds and increase in carbon-to-carbon bonds clearly show a deoxidation of the PI surface, which is in good agreement with the previous analysis of RBS and EDS. The increase in -C-C and -COOH intensities, which is similar to the case of GO, reports rupturing of the benzene rings in the PI structure and can lead to an increase in electrical conductivity.

The C1s signal of untreated PMMA can also be fitted by five overlapping components (Figure 7d): a signal with the highest intensity corresponds to the C-C/C-H bonds in methyl and methylene groups (284.53 eV), a beta carbon peak at 285.20 eV is the response from the -C-C- bonds in C-C=O [58], a -C-OH peak at 286.13 eV is attributed to carbons in the PMMA main chain and to the methyl group that is single-bonded to oxygen. The peaks at 287.28 and 288.88 eV are assigned to the carboxyl/ester groups (-C=O and -COOH) [59,60]. The intensity of carbon bonded to oxygen signals increases at the expense of C-C signals after ion irradiation (Figure 7h), which is probably associated with the breaking of the C-C bonds and the scission of the PMMA main chain [61]. This leads to the active creation of free bonds on the PMMA surface that is subsequently occupied by oxygen from the ambient atmosphere (also confirmed by the RBS and EDS above) [24].

Carbon in the pristine PET is present in three different chemical environments (see Figure 7c): C-C/H (aliphatic/aromatic carbon atoms, 283.35 and 284.77 eV), C-O (methylene carbon atoms, 286.51 eV), and -COO- (ester carbon atoms, 288.87 eV) [62]. The peak corresponding to C=O bonds is not actually observed because it is located and hidden between neighboring large peaks (-COO and -C-OH). After C ion irradiation (Figure 7f): the -C=O peak appears (287.16 eV) along with the current decreasing -COO- and -C-C peaks, and a comparison of data before and after reveals that aromatic carbon mildly decreases relative to the methylene and oxidized carbons [63]. Similar to ERDA, the presence of hydrogen decreases after ion implantation, which is connected to PET chain scission and the release of hydrogen/oxygen molecules, as cited in the SEM part.

### 3.6. Sensing Properties of Prepared Microstructures

The humidity sensing performance of prepared structures was evaluated under various levels of relative humidity (RH) ranging from 5 to 60% in the atmospheric chamber by measurement of the electric sheet resistance of PI, PET, PMMA, and GO, as well as electric capacity changes in GO (see Figure 8). It should be said that the structures prepared in polymers did not show measurable changes in electrical capacity depending on variable humidity. The behavior of the GO microstructure is completely different. Figure 8a shows the almost constant electrical sheet resistivity of GO microstructures depending on the changing humidity, and it is clear that this GO property is unusable for humidity sensing. Figure 8e shows the voltage–frequency characteristic of the circuit shown in Figure 2 (the experimental parts) depending on the relative humidity, with the GO microdevice connected to the circuit as the sensor. One can see that the voltage is a growing function of the humidity of the environment, and thus the capacity of the GO structure increases with the RH level. The GO capacity shifts from the level ~7 pF to ~10 pF following a humidity change from 5 to 60% (see Figure 8f), which implicates the excellent sensoric performance of the GO microstructure. The sheet resistivity of the PI microstructures prepared with carbon fluences of 3.75 × 10^14^ cm^−2^ and 5.625 × 10^14^ cm^−2^ depending on the humidity range (from 5 to 60% RH) is plotted in Figure 8c. The PI microstructure prepared with a carbon fluence 3.75 × 10^14^ cm^−2^ is affected by a decrease in sheet resistivity from the humidity value of 40%. On the contrary, the sheet resistivity of the PI microstructure prepared with a 5.625 × 10^14^ cm^−2^ fluence decreases almost linearly from the lowest to the highest humidity by more than three orders of magnitude. The PI microstructure prepared using the highest ion fluence exhibits an excellent sensoric performance, similar to GO. The electrical sheet resistance of PI microstructure prepared with the highest C fluence decreases almost linearly with the increasing relative humidity. Figure 8b,d show the relationship between the electric sheet resistivity and varying humidity of the micro-structures prepared in PET and PMMA. The resistivity of all the prepared PMMA and PET microstructures decreases with humidity growth, but this decrease is not as significant as in the case of PI. Moreover, the PET and PMMA microstructures have very low sensitivity for low humidity up to 30%. Therefore, there is no evidence of significant trends demonstrating the applicability of these components for humidity sensing. To test for the sensing system’s long-term stability, the same measurements were repeated for structures prepared using the 5.625 × 10^14^ cm^−2^ fluence again after 60 days, when the samples were kept in a room atmosphere in ambient light. It can be seen, in Figure 8b–d, that all the used polymers are quite stable in their behavior and the PI sensing properties are preserved even after a longer time. Unfortunately, in the case of GO, its electrical resistance has significantly decreased and the electrical capacity of the prepared structure on the GO surface has been lost.

## 4. Discussion

In this study, we used the 5 MeV carbon ion-beam writing for the preparation of capacitor-like microstructures with dimensions below 1 mm in GO, PI, PET, and PMMA, and these prepared microstructures were tested for their humidity-sensing properties. In all cases, the carbon ions lost energy and caused damage via electronic stopping, leading to the scission of weak bonds, the release of low-mass gaseous fragments, and the creation of new free bonds. In our cases, the phenyl and benzene rings were breaking, the O and H were released from irradiated areas, while carbonization increased and new six/non-six carbon rings were formed. GO and PI have a better radiation resistance compared to PET and PMMA due to higher aromaticity, and a carbon ion-beam causes less damage in GO and PI compared to PET and PMMA [64]. In PI and GO, there is a minor decrease in O and H concentration, while C concentration increases. On the contrary, the carbon irradiation of PET and PMMA leads to a C and O concentration increase and a very significant decrease in H. Most likely, the atmospheric O bound on the newly irradiation formed bonds in PMMA and PET. The morphology of PI, PET, and GO before and after irradiation is very similar, whereas the PMMA shows very intensive rupturing of surface morphology after ion irradiation, which is probably connected to the low thermal stability of PMMA and ion-induced heating and melting by the high ion-current. The humidity sensing properties of the prepared microstructure were determined by the change in electrical properties depending on various air humidity levels. The best humidity sensing performance was reported in the GO and PI microstructures. In the case of GO, the electric capacity of the micro-sensor increases with the increase in relative humidity. In the case of PI, the sheet resistivity of the prepared micro-sensors decreases with increasing relative humidity. The non-reduced GO in non-irradiated parts is very porous and hydrophilic due to the oxygen groups on the basal planes and edges of GO plates, and water can be adsorbed within the GO layers. Additionally, because the individual GO layers are connected by hydrogen bridges between functional groups and water molecules, the distance between the GO layers increases and the GO structure swells due to the increasing number of hydrogen bonds with increasing relative humidity [10]. Moreover, additional water molecules boost polarization, enhance the dielectric constant, and increase the final electric capacity [2,65]. In the case of PI, the moisture diffuses during adsorption into the polyimide network, where it can either be chemically bound to the oxygen in the ether linkage or between the four carbonyl groups or can condense into micro-voids [66]. The first adsorbed layer of water molecules is stably bonded to the oxides on the polymer surface to form two hydroxyl groups. Further layers of water molecules then become disorderly bound by weak van der Waals forces to the first layer, and a Grotthus mechanism occurs to release a proton from one oxygen atom (in the water molecule) to another [5]. This treatment of water molecules in the polyimide network then causes conductivity to increase [66]. In addition, the ion irradiation of GO and PI increases the number of free bonds, which enables better interaction with water vapor and leads to an increase in the volume of bounded water. On the other hand, the free bonds that arise in PET and PMMA after carbon irradiation are occupied by oxygen from the atmosphere after the sample is transferred from a vacuum and significantly limit the interaction of irradiated PET and PMMA with water molecules, especially for low values of relative humidity.

## 5. Conclusions

We reported the fabrication of novel resistive and capacitive types of humidity sensors based on a capacitor-like microstructure prepared by 5 MeV carbon micro-beam writing in GO and PI substrates. The prepared structures are a system of 10 µm-wide electrodes with a mutual mean distance of 20 µm; the total dimension of the structures is 900 × 900 µm^2^. Sensoric performance was tested in an atmospheric chamber with a relative humidity (RH) varying from 5 to 60% and by measuring the change in electrical sheet resistivity and capacity depending on relative humidity. In general, we demonstrated the great potential of carbonion micro-lithography for preparing humidity sensors in GO and PI substrates, which are able to operate over a broad humidity range with very good sensitivity, and have high potential in the implementation of practical humidity sensors.

## Figures and Tables

**Figure 1 polymers-15-01066-f001:**
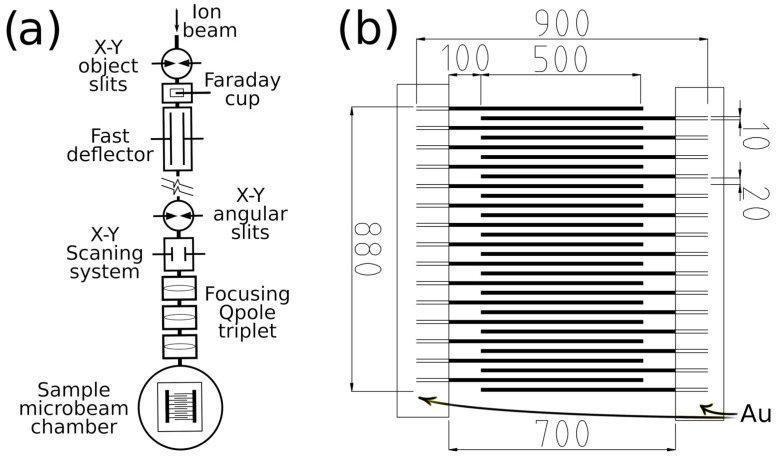
(**a**) A schematic illustration of the maskless production of a micro-structure via ion-beam writing. (**b**) A schematic illustration of the sensory microstructure prepared by carbon ion-beam writing on the surfaces of GO, PET, PI, and PMMA.

**Figure 2 polymers-15-01066-f002:**
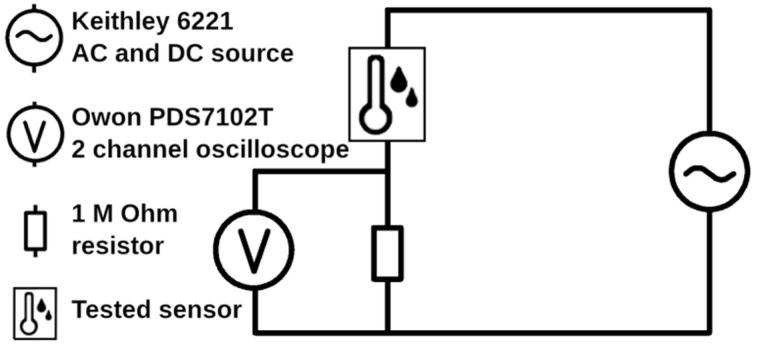
An equivalent scheme (high-pass filter) as was used to measure the change in the capacitance of the prepared micro-structures with varying humidity.

**Figure 3 polymers-15-01066-f003:**
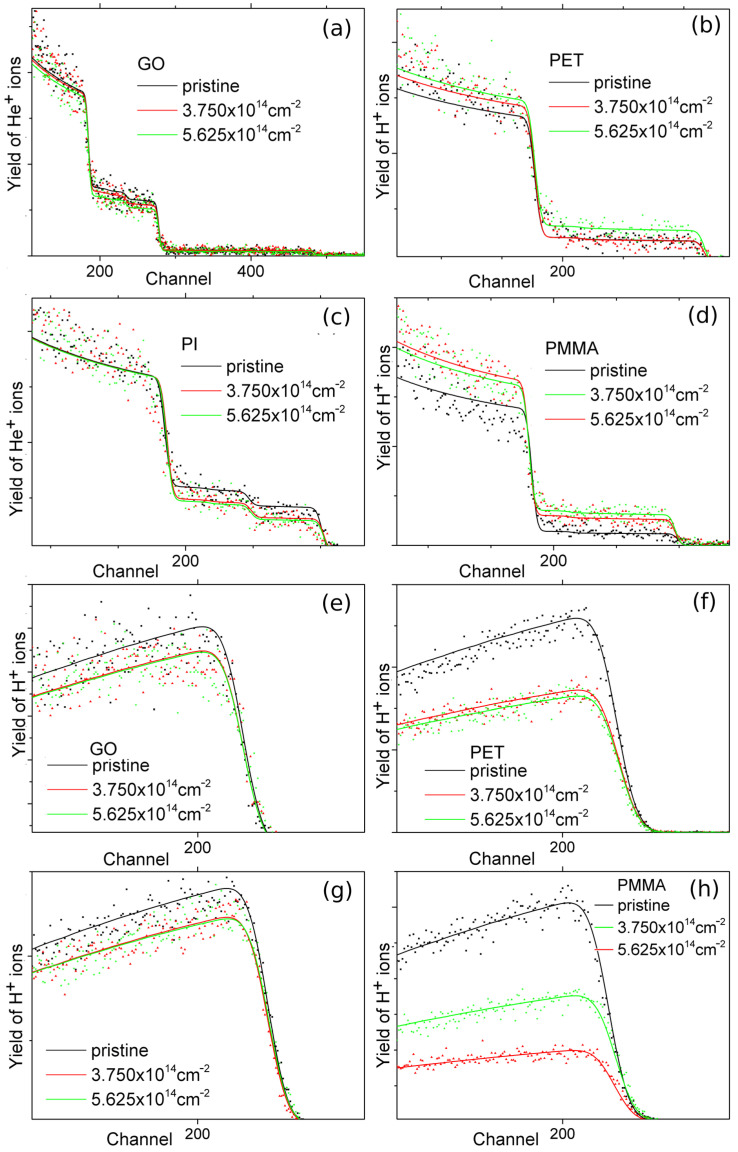
The RBS experimental spectra (points) and SIMNRA simulated spectrum (line) for GO (**a**), PET (**b**), PI (**c**), and PMMA (**d**) and the experimental ERDA (points) and SIMNRA simulated (line) spectra for GO (**e**), PET (**f**), PI (**g**), and PMMA (**h**), all pristine and irradiated by 5.0 MeV carbon ions with fluences of 3.75 × 10^14^ cm^−2^ and 5.625 × 10^14^ cm^−2^.

**Figure 4 polymers-15-01066-f004:**
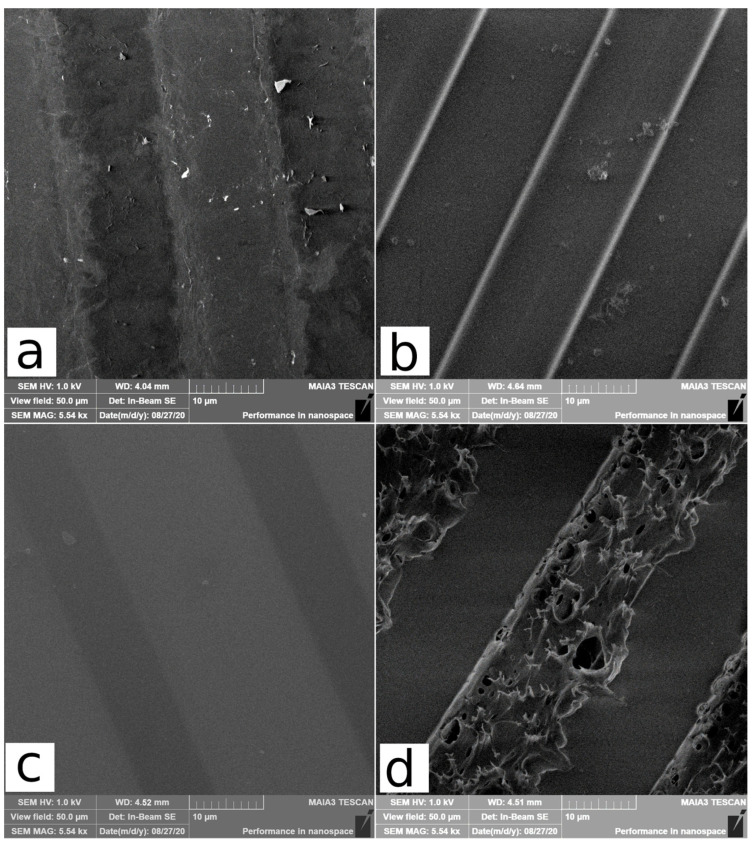
The SEM images of the structures prepared using ion-beam writing on surfaces of GO (**a**), PET (**b**), PI (**c**), and PMMA (**d**). These samples were irradiated with a fluence 5.625 × 10^14^ cm^−2^.

**Figure 5 polymers-15-01066-f005:**
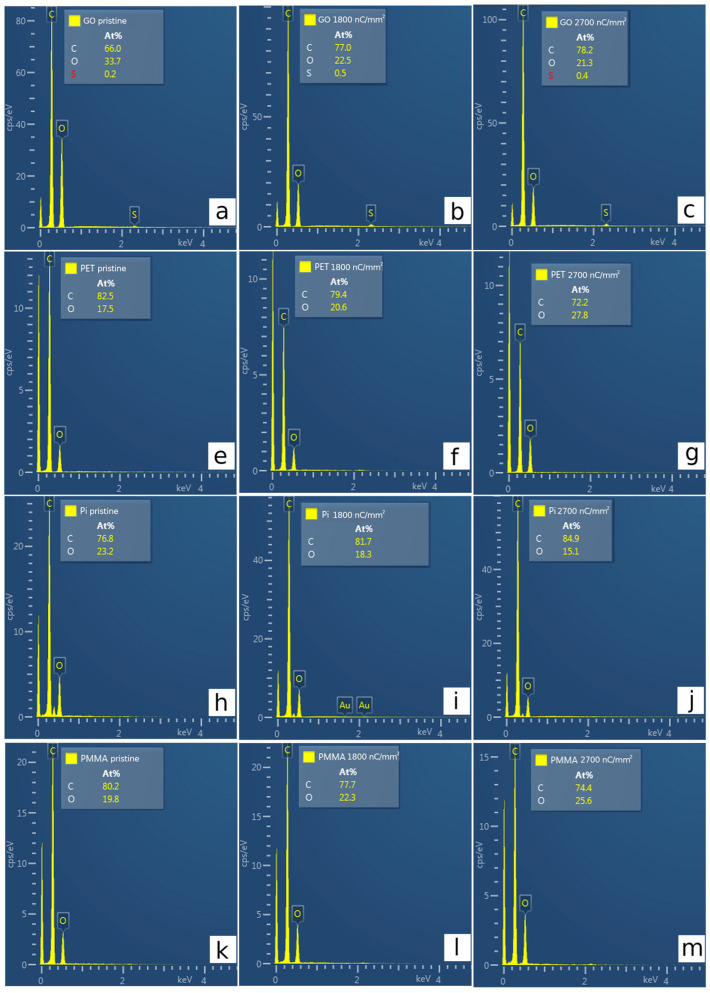
The EDS spectra of unaffected and irradiated foils of GO (**a**–**c**), PET (**e**–**g**), PI (**h**–**j**), and PMMA (**k**–**m**). The fluence 1800 nC/mm^2^ equals 3.75 × 10^14^ cm^−2^ and 2700 nC/mm^2^ equals 5.625 × 10^14^ cm^−2^.

**Figure 6 polymers-15-01066-f006:**
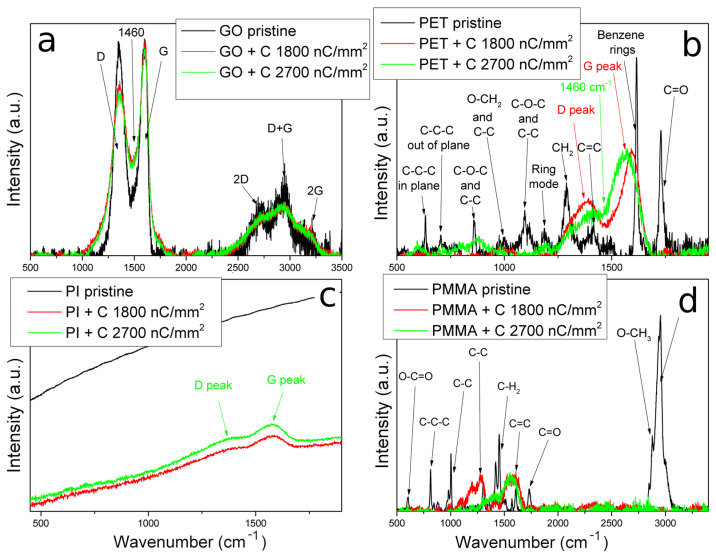
The Raman spectra of pristine and irradiated GO (**a**), PET (**b**), PI (**c**), and PMMA (**d**). The fluence 1800 nC/mm^2^ equals 3.75 × 10^14^ cm^−2^ and 2700 nC/mm^2^ equals 5.625 × 10^14^ cm^−2^.

**Figure 7 polymers-15-01066-f007:**
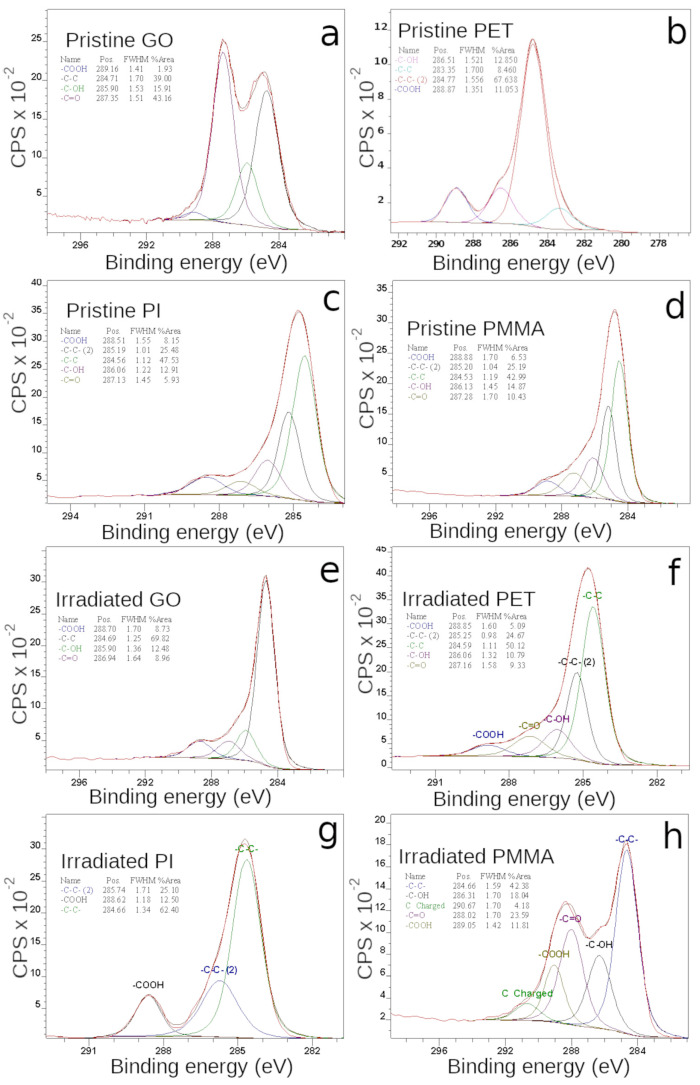
The deconvoluted C1s peaks from high-resolution XPS spectra of pristine and irradiated GO (**a**,**e**), PET (**b**,**f**), PI (**c**,**g**), PMMA (**d**,**h**). The irradiation was performed with fluence 5.625 × 10^14^ cm^−2^.

**Figure 8 polymers-15-01066-f008:**
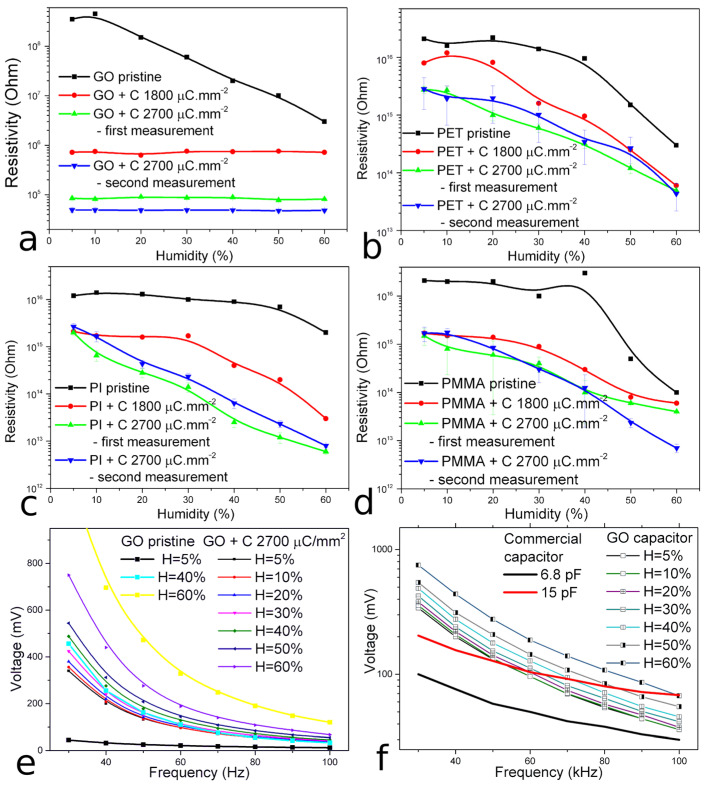
The electric resistivity and frequency–voltage characteristics of prepared micro-structures depending on the relative humidity: (**a**) The sheet resistivity of GO microstructures, (**b**) the sheet resistivity of PET microstructures, (**c**) the sheet resistivity of PI microstructures, (**d**) the sheet resistivity of PMMA microstructures, (**e**) the voltage–frequency characteristic of the structure prepared on the GO surface with a fluence 5.625 × 10^14^ cm^−2^, (**f**) the voltage–frequency characteristic of the structure prepared on the GO surface with a fluence of 5.625 × 10^14^ cm^−2^ in logarithmic scale and compared with commercial ceramic capacitors.

**Table 1 polymers-15-01066-t001:** The GO, PET, PI, and PMMA elemental composition before and after irradiation with carbon ions determined using RBS and ERDA.

	C (at. %)	O (at. %)	H (at. %)	N (at. %)	C/O	C/H
PET pristine	57.7 ± 1.9	6.0 ± 0.4	36.3 ± 2.3		9.6	1.6
PET 3.750 × 10^14^ cm^−2^	67.4 ± 2.2	6.4 ± 0.4	26.2 ± 1.7		10.5	2.6
PET 5.625 × 10^14^ cm^−2^	65.2 ± 2.1	9.5 ± 0.5	25.4 ± 1.6		6.9	2.6
PI pristine	56.4 ± 1.8	12.8 ± 0.7	25.6 ± 1.7	5.2 ± 0.3	4.4	2.2
PI 3.750 × 10^14^ cm^−2^	61.7 ± 2.0	9.5 ± 0.6	23.6 ± 1.6	5.2 ± 0.3	6.5	2.6
PI 5.625 × 10^14^ cm^−2^	62.4 ± 2.0	9.2 ± 0.5	23.0 ± 1.5	5.4 ± 0.3	6.8	2.7
PMMA pristine	54.2 ± 1.8	2.7 ± 0.3	43.0 ± 2.8		20.0	1.3
PMMA 3.750 × 10^14^ cm^−2^	62.6 ± 2.0	8.3 ± 0.8	29.0 ± 1.9		7.5	2.2
PMMA 5.625 × 10^14^ cm^−2^	74.2 ± 2.4	7.6 ± 0.7	18.0 ± 1.2		9.8	4.1
GO pristine	68.0 ± 2.5	19.0 ± 1.0	9.0 ± 0.7	3.0 ± 0.2	3.6	7.6
GO 3.750 × 10^14^ cm^−2^	70.0 ± 2.6	18.0 ± 0.9	8.0 ± 0.6	4.0 ± 0.4	3.9	8.8
GO 5.625 × 10^14^ cm^−2^	71.0 ± 2.6	16.0 ± 0.8	8.0 ± 0.6	3.0 ± 0.2	4.4	8.9

## Data Availability

Not applicable.

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
