# Peer review of "Graphene Oxide and Polymer Humidity Micro-Sensors Prepared by Carbon Beam Writing"

_polymers, 2023, doi:10.3390/polym15051066_

Round 1

Reviewer 1 Report

Petr Malinský et al. reported the graphene oxide and polymers humidity micro-sensors prepared by carbon beam writing. The authors have characterized the sample extensively to achieve properties that directly affect the humidity sensing. However, a few more clarifications are needed before the manuscript can be considered suitable for publication. A list of other comments that need to be addressed is as follows:

1.     The abstract needs to be modified to eliminate vague sentences, and replace with actual significant data and findings.

2.     Abstract: What is the reason for better humidity responses? Is it synergistic effect or structural advantage?

3.     An introduction is poorly written; please include the scope and limitations of the graphene oxide and polymers-based humidity sensing? Methods used for humidity sensing. Why authors choose this technique? Please justify the novelty.

4.     Abstract: important information missing, selectivity compared to other interference/analytes, and long-term stability.

5.     The authors should discuss existing methods used for humidity sensing

6.     It is better to mention in the introduction how the selectivity could be enhanced?

7.     The quality of some figures is inferior and needs to be enhanced.

8.     It is better to check and correct the font size of the x and y-axis of all figures in the manuscript. It should be the same.

9.     Selectivity plot missing with other analytes.

10.  Please repeat the data with lower to higher concentrations of humidity.

11.  Please cite some references to justify the humidity sensing with 2D materials and justify the novelty: ACS Nano 2013, 7, 12, 11166–11173, ACS Appl. Mater. Interfaces 2019, 11, 1, 1699–1705, Adv. Funct. Mater. 2022, 32, 2107330, Journal of Nanoscience and Nanotechnology, Volume 19, Number 8, August 2019, pp. 5310-5316(7), Scientific Reports volume 3, Article number: 2714 (2013)

Author Response

  1. The abstract needs to be modified to eliminate vague sentences, and replace with actual significant data and findings.

The abstract has been completely rewritten according to the recommendation.

  1. Abstract: What is the reason for better humidity responses? Is it synergistic effect or structural advantage?

A sentence has been added to the abstract: “The two low fluences (the 3.75×1014 cm-2 and 5.625×1014 cm-2) of carbon ions with energy of 5 MeV were used, where we assume mainly the influence of structural changes in the irradiated materials.”

  1. An introduction is poorly written; please include the scope and limitations of the graphene oxide and polymers-based humidity sensing? Methods used for humidity sensing. Why authors choose this technique? Please justify the novelty.

The introduction has been expanded and supplemented with new information from more recent publications. Some other types of humidity sensors have been added with ranges and limitations. The advantages of ion lithography have been mentioned.

  1. Abstract: important information missing, selectivity compared to other interference/analytes, and long-term stability.

The abstract has been completely rewritten as recommended.

  1. The authors should discuss existing methods used for humidity sensing.

Some other types of humidity sensors were added into the introduction and discussed.

  1. It is better to mention in the introduction how the selectivity could be enhanced?

The introduction has been modified only for the humidity sensors.  

  1. The quality of some figures is inferior and needs to be enhanced.

 The quality of figures was enhanced

  1. It is better to check and correct the font size of the x and y-axis of all figures in the manuscript. It should be the same.

The fontsize in the figures was enlarged

  1. Selectivity plot missing with other analytes.

The article focuses only on humidity sensors

  1. Please repeat the data with lower to higher concentrations of humidity.

The measurement was repeated and data about time stability added.

  1. Please cite some references to justify the humidity sensing with 2D materials and justify the novelty: ACS Nano 2013, 7, 12, 11166–11173, ACS Appl. Mater. Interfaces 2019, 11, 1, 1699–1705, Adv. Funct. Mater. 2022, 32, 2107330, Journal of Nanoscience and Nanotechnology, Volume 19, Number 8, August 2019, pp. 5310-5316(7), Scientific Reports volume 3, Article number: 2714 (2013)

The introduction and discussion has been expanded and supplemented with recommended publications.

Reviewer 2 Report

The authors report the fabrication of the micro-scale humidity sensors based on carbon ion beam writing on graphene oxide (GO) and polymeric films. The authors have represented the detailed characterization studies of the polymeric film and the GO using the ion fluencies of 3.75´1014 and 5.625´1014 cm-2. The manuscript has significant and valuable scientific information to be considered for publication in Polymers. However, the authors need to respond to some questions before publication.

1.     Please schematically represent the fabrication of carbon ion beam writing for the maskless production of an electrode system for humidity sensing applications.

2.     From Table 1, on comparison of PMMA pristine to the irradiated PMMA samples at 3.75´1014 and 5.625´1014 cm-2, why the atomic % of C, H, and O have drastic differences?

3.     Does carbon ion irradiation increase the atomic % of C? If not, why do the PI and GO show increments in the atomic % compared to the PET and PMMA?

4.     In lines 175-176, the author mentioned that the growth of O concentration in PET
and PMMA after ion irradiation can be caused by the post-implantation oxidation of the
radiation-damaged layer. Why is there no consistency in the C/O ratio of PET and the high C/O ratio of PMMA pristine?

5.      The scale bar of the SEM image of Figure. 3d is 20 µm, but others are 10 µm. It is suggested to present the SEM images with a consistent scale bar.

6.     Please explain why the pristine GO shows a trace amount of Sulphur in the EDX analysis in Figure 5(a-c)?

7.  Why does the PI samples in Figure 5i is presented with Au peak? Do the SEM images of the GO and polymeric samples are recorded after the Au sputtering? If so, why didn’t the other EDX analysis doesn’t show any Au peak?

8.  It is recommended to the authors to follow one particular order as GO (a), PET (b), PI (c) and PMMA (d) in the Figure arrangement. It will be beneficial for the readers.

9.  Please try to maintain the same notation of units for ion fluencies.

Author Response

  1. Please schematically represent the fabrication of carbon ion beam writing for the maskless production of an electrode system for humidity sensing applications.

The schematic illustration of the micro-beam writing was added into Fig. 1.

  1. From Table 1, on comparison of PMMA pristine to the irradiated PMMA samples at 3.75´1014and 5.625´1014 cm-2, why the atomic % of C, H, and O have drastic differences?

We checked the RBS and ERDA results for the PMMA irradiated samples and it turned out that the results are swapped. The table and figure have been corrected.

  1. Does carbon ion irradiation increase the atomic % of C? If not, why do the PI and GO show increments in the atomic % compared to the PET and PMMA?

The sentence “The irradiation with 5 MeV carbon ions with predominant electronic stopping leads to the weak bond scission in GO and PI and release of low-mas gaseous fragments, followed by GO and PI reduction and carbonization.” was added.

  1. In lines 175-176, the author mentioned that the growth of O concentration in PET
    and PMMA after ion irradiation can be caused by the post-implantation oxidation of the
    radiation-damaged layer. Why is there no consistency in the C/O ratio of PET and the high C/O ratio of PMMA pristine?

PET and PMMA have completely different structure, composition and radiation/heat resistance and thus they have very different behaviour.

  1. The scale bar of the SEM image of Figure. 3d is 20 µm, but others are 10 µm. It is suggested to present the SEM images with a consistent scale bar.

The Fig. 3D was changed.

  1. Please explain why the pristine GO shows a trace amount of Sulphur in the EDX analysis in Figure 5(a-c)?

 The sentence “The trace amounts of sulfur in the GO originate from GO film synthesis (see Fig. 5a-c).” was added into the Results chapter.

  1. Why does the PI samples in Figure 5i is presented with Au peak? Do the SEM images of the GO and polymeric samples are recorded after the Au sputtering? If so, why didn’t the other EDX analysis doesn’t show any Au peak?

The micro-capacitor was prepared between two Au contacts (please see on Fig. 1) and this signal is probably reflection of the trace amount of gold on part of surface.

  1. It is recommended to the authors to follow one particular order as GO (a), PET (b), PI (c) and PMMA (d) in the Figure arrangement. It will be beneficial for the readers.

The figures were corrected.

  1. Please try to maintain the same notation of units for ion fluencies.

The text of the paper was corrected and explanations were added to the images for better clarity.

Reviewer 3 Report

I consider that the authors of the manuscript "The graphene oxide and polymers humidity micro-sensors prepared by carbon beam writing" have done a good job and the current version is ready to be published.

Author Response

Thank the reviewer for the revision.

Reviewer 4 Report

This paper deals with the study of micrometer-scale electrode systems in insulating matrixes for humidity sensors application, based on carbon ion beam writing. In my opinion, this is a very intriguing approach employing a preparation method, by ion irradiation, which is a quite interesting and well-discussed technique for chemical and functional property modification. The properties of the samples have been studied in a very thorough and detailed way, and the results appear very interesting. The findings will have a positive impact on the development of resistive and capacitive types of humidity sensors and various engineering applications. The conclusions are quite relevant. So, minor aspects should be revised before the publication of this manuscript in Polymers.

1)      English may be polished in the revision.

2)      Some references about this topic may be read during the revision and might be cited in the introduction part of the revision to extend the readership.

Author Response

1)      English may be polished in the revision.

The paper has undergone language correction

2)      Some references about this topic may be read during the revision and might be cited in the introduction part of the revision to extend the readership.

The introduction has been expanded and supplemented with new information from more recent publications.

Round 2

Reviewer 1 Report

All the comments are addressed properly, the manuscript is ready for the publication

Reviewer 2 Report

I think, authors are revised well. Therefore, Current form of the manuscript has been accepted for publication.